# Short-Term Growth Hormone Administration Mediates Hepatic Fatty Acid Uptake and De Novo Lipogenesis Gene Expression in Obese Rats

**DOI:** 10.3390/biomedicines11041050

**Published:** 2023-03-29

**Authors:** Sutharinee Likitnukul, Sumpun Thammacharoen, Orada Sriwatananukulkit, Chanathip Duangtha, Ruedee Hemstapat, Chotchanit Sunrat, Supachoke Mangmool, Darawan Pinthong

**Affiliations:** 1Department of Pharmacology, Faculty of Science, Mahidol University, Bangkok 10400, Thailand; 2Department of Physiology, Faculty of Veterinary Science, Chulalongkorn University, Bangkok 10330, Thailand; 3Department of Physiology, Faculty of Science, Mahidol University, Bangkok 10400, Thailand

**Keywords:** GH, DIO rats, hepatic lipid metabolism, NAFLD, insulin resistance

## Abstract

Obesity has been linked to metabolic syndrome, type 2 diabetes, and non-alcoholic fatty liver disease (NAFLD). Obesity causes a decrease in growth hormone (GH) levels and an increase in insulin levels. Long-term GH treatment increased lipolytic activity as opposed to decreasing insulin sensitivity. Nonetheless, it is possible that short-term GH administration had no impact on insulin sensitivity. In this study, the effect of short-term GH administration on liver lipid metabolism and the effector molecules of GH and insulin receptors were investigated in diet-induced obesity (DIO) rats. Recombinant human GH (1 mg/kg) was then administered for 3 days. Livers were collected to determine the hepatic mRNA expression and protein levels involved in lipid metabolism. The expression of GH and insulin receptor effector proteins was investigated. In DIO rats, short-term GH administration significantly reduced hepatic fatty acid synthase (FASN) and cluster of differentiation 36 (CD36) mRNA expression while increasing carnitine palmitoyltransferase 1A (CPT1A) mRNA expression. Short-term GH administration reduced hepatic FAS protein levels and downregulated gene transcription of hepatic fatty acid uptake and lipogenesis, while increasing fatty acid oxidation in DIO rats. DIO rats had lower hepatic JAK2 protein levels but higher IRS-1 levels than control rats due to hyperinsulinemia. Our findings suggest that short-term GH supplementation improves liver lipid metabolism and may slow the progression of NAFLD, where GH acts as the transcriptional regulator of related genes.

## 1. Introduction

Obesity is a major global public health issue. The prevalence of obesity has continued to increase in the past few decades [1,2]. Obesity potentiates multiple negative health effects, including cardiovascular disease, metabolic syndrome, diabetes mellitus, as well as non-alcoholic fatty liver disease (NAFLD) [3,4]. Furthermore, changes in hormonal profiles have been observed in the obesity state, including decreased growth hormone (GH) levels [5,6], and increased insulin and leptin levels [7,8]. These hormonal changes lead to remarkable alterations in lipid and glucose metabolism. GH, in particular, has a significant lipolytic effect and potentiates elevated glucose levels, which appears to counteract the effects of insulin [9]. Supplementation of GH for long periods, especially longer than 21 days, significantly reduced fat accumulation in rats [10,11]. However, long-term GH treatment promoted insulin resistance in hypophysectomized animals and GH-deficient subjects [12,13]. On the other hand, short-term GH treatment (5 days) did not increase plasma insulin levels in obese animals [14] and did not affect fat accumulation [15]. In addition, short-term GH treatment (5 days) promoted the anorectic effect of obese rats [14]. This anorectic effect was intriguing since it would offer a useful method of body weight management. This treatment mimics the physiological condition of increased GH secretion, especially during exercise [16,17]. In humans, the effects of GH supplementation on body adiposity varied depending on the duration of treatment. For instance, patients with GH deficiency (GHD) were found to benefit from GH supplementation for the first few months, while long-term supplementation increased subcutaneous fat [18].

Physical inactivity and eating a hypercaloric diet are the primary causes of weight gain, eventually leading to obesity [19]. Imbalances in energy status that occur in conjunction with obesity, including insulin resistance and fat accumulation in the liver, contribute to the development of NAFLD. In addition, the elevation of hepatic triglycerides (TG) is one of the hallmarks of NAFLD [4,20] and increased lipogenesis has been reported in NAFLD patients [4]. The imbalance of the genes involved in lipid synthesis and disposal leads to the progression of NAFLD [4]. To maintain physiologic conditions, it is crucial to regulate the transcription of genes involved in liver lipid synthesis and storage, including fatty acid synthase (FASN), acetyl-CoA carboxylase (ACC), and cluster of differentiation 36 (CD36), and the genes involved in lipid disposals such as carnitine palmitoyltransferase 1A (CPT1A) and the very low-density lipoprotein receptor (VLDLR). Hepatic lipogenesis is regulated by hormonal and nutritional status with GH and insulin reported to regulate hepatic gene transcription [21,22]. Hepatic CD36 mRNA expression was higher in GH receptor (GHR) knockout mice and liver-specific GHR knockout mice, which was linked to higher levels of hepatic TG accumulation [23,24]. Additionally, hepatic FASN mRNA expression was greater in liver-GHR knockdown mice compared to control mice after fasting and refeeding. These findings confirmed the importance of hepatic GHR in liver lipid metabolism, especially in de novo lipogenesis [25]. Insulin is involved in hepatic lipogenesis by upregulating the expression of FASN mRNA in diabetic and transgenic mice [26]. Insulin resistance is common in patients with NAFLD [27], and it has been established that insulin resistance is one of the major pathological factors in disease progression [28]. Reduced GH levels may contribute to the advancement of NAFLD, although the roles of GH in lipid metabolism are complex. Short-term GH supplements may reduce hepatic insulin resistance by improving liver lipid metabolism.

To investigate whether short-term GH administration could affect hepatic lipid metabolism, the diet-induced obesity (DIO) rat model was used. Consuming a hypercaloric diet induces obesity and NAFLD, which subsequently decreases GH levels and insulin resistance [29,30,31]. Lipid accumulation in the liver has been found after feeding with a hypercaloric diet [29,32,33,34]. Feeding rats with a high-fat diet decreased circulating GH levels and the downstream effectors of the hepatic GH signaling pathway, including signal transducer and activator of transcription-5 (STAT-5) and Janus kinase-2 (JAK2) [31]. Hepatic JAK2 deletion mice develop a fatty liver that is associated with an elevated level of CD36 gene expression [35]. GHR also plays an important role in liver lipid metabolism. Increased hepatic lipid uptake, increased hepatic de novo lipogenesis, and severe insulin resistance were observed in the livers of GHR knockout mice [36]. In addition, increased hepatic triglyceride content was found in the GHR antagonist transgenic mouse model [37]. Thus, the regulation of GHR may affect the severity of NAFLD.

Supplementation of GH may have the potential to reduce the progression of NAFLD in obese rats. We hypothesized that short-term GH administration could provide benefits for either regulating the hepatic gene transcription related to lipid metabolism or decreasing hepatic lipid uptake and accumulation in the liver. Moreover, we investigated the underlying mechanisms through which GH affects hepatic lipid metabolism. Our findings support the potential mechanisms of short-term recombinant human GH for further study in the treatment of NAFLD.

## 2. Materials and Methods

### 2.1. Animals and Diet

Forty-seven male Wistar rats, aged 12 weeks, from the National Laboratory Animal Center at Mahidol University, were used for this study. The study protocol was approved by the Institutional Animal Care and Use Committee (IACUC), Faculty of Veterinary Science, Chulalongkorn University (approval protocol number #1531026). Animals were individually housed in conventional hanging cages with stainless steel wire mesh floors (33 cm × 18 cm × 20 cm) under standard conditions in a temperature-controlled room (12 h/12 h light/dark cycle, 22 ± 1 °C). The rats were allowed to access water ad libitum and standard rat chow (#082; Perfect Companion Group Ltd., Samutprakarn, Thailand) containing protein 24 g%, carbohydrate 42 g%, fat 4.5 g%, energy 3.04 kcal/g, and energy from fat 13%. After acclimatization for 2 weeks, they were divided into two main groups. The control group was fed with the standard rat chow (*n* = 12) and the hypercaloric diet (HC diet) feeding group was given an HC diet (protein 17.52%, carbohydrate 30.66%, fat 30.3%, energy 4.65 kcal/g, energy from fat 60%; *n* = 35) for 6 weeks. The HC diet was made by combining melting lard and standard rat chow, with a lard/chow ratio of 27 and 73 g%, respectively. At 20 weeks old, HC-fed rats were arranged by increasing body weight (BW); the lowest tertile of BW gainers were identified as diet-resistant (DR) rats (*n* = 10) which showed a similar final BW as control rats and no obesity was observed. These rats were represented as the group of HC diet control. Diet-induced obesity (DIO) rats (*n* = 12) showed a higher BW gain, final BW, and body fat mass compared to control rats. The middle BW gainers (*n* = 13), which did not meet the DR or DIO criteria, were excluded for this experiment. The timeframe of the experiment and the groups of HC-feeding rats are represented in Figure 1a,b, respectively. Then an intraperitoneal glucose tolerance test (IPGTT) was undertaken (see below). The measurements of BW and food intake (FI) (23 h FI, ±0.1 g corrected for spillage) were recorded during the experimental period.

The control, DIO, and DR rats were randomly assigned to treat with either GH or normal saline (NSS) as vehicle. The experimental groups were described as follows:

Group 1: Control rats + NSS (*n* = 6)

Group 2: Control rats + GH (*n* = 6)

Group 3: DIO rats + NSS (*n* = 6)

Group 4: DIO rats + GH (*n* = 6)

Group 5: DR rats + NSS (*n* = 5)

Group 6: DR rats + GH (*n* = 5)

The GH-treated rats were subcutaneously (s.c.) injected with GH (GenHeal^®^; Shanghai United Cell Biotechnology Co., Ltd., Shanghai, China) at 1 mg/kg twice daily (08:00 h and 16:00 h) for 3 days, whereas other subgroups received NSS injections, as shown in Figure 2.

After the short-term GH administration, the rats were fasted overnight and sacrificed using pentobarbital (150 mg/kg, Nembutal^®^; Ceva Sante Animale, Libourne, France). Rats were then transcardially perfused with 0.1 M phosphate-buffered saline (PBS). To determine total body fat, the visceral and subcutaneous adipose tissues, including mesenteric, retroperitoneal and perirenal, epididymal, interscapular and inguinal fat pads, were dissected and weighted. The liver was collected for triglyceride (TG) measurement, hepatic gene expression, and effector protein determination.

### 2.2. Intraperitoneal Glucose Tolerance Test

An intraperitoneal glucose tolerance test (IPGTT) was performed to measure systemic insulin sensitivity. After a 6-week feeding period with HC or a control diet, the rats were fasted overnight (16 h). Basal blood samples were collected by the tail-clipping technique and fasting blood glucose was measured using a blood glucose meter (Accu-Chek^®^ Performa; Roche Diagnostic GmbH, Mannheim, Germany). Glucose administration (2 g/kg; 50% glucose solution; A.N.B. Laboratories Co., Ltd., Bangkok, Thailand) was performed via intraperitoneal (i.p.) injection. Blood samples were collected for blood glucose level measurement after 15, 30, 60, 90, and 120 min following glucose injection.

### 2.3. Calculation of Homeostasis Model of Insulin Resistance (HOMA-IR)

Blood samples were collected from the ventral tail artery after a 16-h food withdrawal period. Samples were centrifuged at 3000× *g*, 4 °C for 15 min to obtain the plasma and stored at −20 °C until the analysis was performed. Plasma insulin was measured using an insulin ELISA kit (EZRMI-13K; Merck Millipore, MA, USA) according to the instructions of the manufacturer.

The index of HOMA-IR was calculated from fasting glucose and fasting insulin levels as per formula:

HOMA-IR = (fasting glucose (mg/dL) × fasting insulin (µU/mL))/2430 [38].

### 2.4. Measurement of Circulating Free Fatty Acids, Total Cholesterol and Triglyceride Levels

Estimation of free fatty acid (FFA) levels was performed from plasma as described in Section 2.3. A commercially colorimetric FFA assay kit (ab65341; Abcam, Cambridge, UK) was used to determine FFA levels. Total cholesterol and triglyceride levels were estimated by using the serum. In this regard, blood samples were collected from the heart before the transcardial perfusion was made. Then the blood samples were centrifuged at 3000× *g*, 4 °C for 15 min to obtain the serum and stored at −20 °C until the analysis was conducted. A chemistry analyzer system (Beckman Coulter AU400, Brea, CA, USA) was used to determine serum total cholesterol and triglyceride levels by enzymatic methods.

### 2.5. Liver Triglyceride (TG) Measurements

Livers were collected after euthanasia with pentobarbital i.p. injection (150 mg/kg, Nembutal^®^; Ceva Sante Animale, Libourne, France) and perfusion with 0.1 M PBS. Livers were immediately frozen on dry ice and kept at −80 °C. Liver tissue samples (100 mg) were homogenized in 1 mL of 5% NonidetTM P40 Substitute (NP40; #74385; Sigma-Aldrich^®^, St. Louis, MO, USA) as described by Huang et al., 2020 [39]. Samples were then heated to 80–100 °C for 2–5 min and cooled to room temperature with this step repeated once. TG concentration was measured using a triglyceride quantification kit (#MAK266; Sigma-Aldrich, St. Louis, MO, USA) following the manufacturer’s instructions.

### 2.6. RNA Isolation and Quantitative RT-PCR

Liver RNA was extracted using the TRIzol™ reagent (Invitrogen, Carlsbad, CA, USA). Briefly, liver tissue weighing 80 mg was homogenized in 1 mL of TRIzol™ reagent. Phase separation was performed using chloroform (200 μL). Following centrifugation at 12,000× *g* at 4 °C for 15 min, the supernatant was collected. Isopropanol (500 µL) was added to the samples to precipitate the RNA and then centrifuged at 12,000× *g* for 10 min at 4 °C. The supernatant was discarded and the RNA pellet was washed with ice-cold 75% ethanol. Samples were centrifuged at 7500× *g*, 4 °C for 5 min. The supernatant was discarded, and the RNA pellet was allowed to air dry. RNA was redissolved in DNase/RNase-free distilled water (100 μL) (Invitrogen, Carlsbad, CA, USA), and RNA purity was determined using the Nanodrop 2000C. (Thermo Scientific, Waltham, MA, USA). Synthesis of cDNA utilized the iScript™ cDNA synthesis kit (Hercules, CA, USA) according to the manufacturer’s instructions. Target gene amplification was detected using SYBR Green (Luna^®^ Universal qPCR Master Mix, New England Biolabs, Ipswich, MA, USA) and specific sequence primers were designed using NCBI/Primer-Blast (Table 1). To monitor DNA amplification, samples were analyzed using an ABI PRISM7500 Sequence Detection System with analytical software (Applied Biosystems, Carlsbad, CA, USA). The expression levels of gene transcription were normalized with β-actin.

### 2.7. Western Blotting

Frozen liver tissue was homogenized in an ice-cold RIPA buffer (Cell Signaling Technology, Danvers, MA, USA) containing protease inhibitors (Cell Signaling Technology, Danvers, MA, USA) and phosphatase inhibitors (PhosSTOP™, Merck Millipore, Darmstadt, Germany). Homogenization utilized a Dounce glass tissue homogenizer (Kimble^®^ #D8938; Sigma-Aldrich®, Munich, Germany). Samples were sonicated and centrifuged at 8000× *g* at 4 °C for 5 min to remove cell debris. The protein concentration of the supernatant (beneath the lipid layer) was measured using the BCA protein assay kit (Merck Millipore, Darmstadt, Germany). Proteins were separated using 10% sodium dodecyl sulfate-polyacrylamide gel electrophoresis (SDS-PAGE) and transferred to a polyvinylidene fluoride (PVDF) membrane. The membrane was blocked from non-specific binding using 5% non-fat dry milk and incubated overnight at 4 °C with primary antibodies: β-actin #4970, phosphatase and tensin homolog (PTEN #9188), phosphatidylinositol-3-kinase p110 subunit alpha (PI3K p110 α #4249), JAK2 #3230, STAT5 #94205, AKT #9272, insulin receptor substrate-1 (IRS-1 #2390), fatty acid synthase (FAS #3180) (Cell Signaling Technology^®^, Danvers, MA, USA). After washing, the membrane was incubated with peroxidase-conjugated IgG fraction monoclonal mouse anti-rabbit IgG (#211-031-171; Jackson Immunoresearch laboratory^®^, West Grove, PA, USA). The specific protein bands were developed by incubation for 5 min with ECL substrate (Immobilon^®^ Crescendo Western HRP Substrate; #WBLUR0500; EMD Millipore, Burlington, MA, USA). Membranes were imaged with Amersham HyperfilmTM (GE Healthcare, Buckinghamshire, UK) and the densitometry was analyzed using ImageJ software (NIH, Bethesda, MA, USA). The results were normalized with the densitometry of β-actin.

### 2.8. Statistical Analysis

Data were tested for normality based on the Shapiro–Wilk test. Results are presented as mean ± standard error of the mean (SEM). Significant different values were considered as *p* ≤ 0.05. Analysis of significant mean difference was performed using one-way analysis of variance (ANOVA) for the difference between groups. Analysis of the effect of GH administration and the difference among groups was assessed by two-way ANOVA. A Bonferroni test was used for post hoc analysis. In addition, analysis of the gene between GH and saline-treated groups utilized an independent T-test. Data analysis was performed using GraphPad Prism 7.00 (GraphPad Software, San Diego, CA, USA).

## 3. Results

### 3.1. Characteristics of Diet-Induced Obesity (DIO) and Diet-Resistant (DR) Rats

After the obesity induction period (6 weeks of HC diet feeding), the HC rats with the highest tertile of BW gain were selected as DIO rats (*n* = 12). HC rats with a final BW similar to control groups were selected as DR rats (*n* = 10). The BW of each group is presented in Figure 3a. The FI measurements were used to determine energy consumption per day. The FI results indicated that DIO rats had higher energy intake relative to DR and control rats throughout the period of HC feeding. However, energy intake in DR rats was higher compared to control rats (Figure 3b). The percentage of body fat mass was significantly different in each group, with the highest fat mass seen in DIO rats, followed by DR and control rats, respectively (Figure 3c). The results of each specific site of adipose tissue were shown in the supplementary materials (Appendix A). The DIO rats also exhibited the highest BW gain, followed by DR and control rats, respectively (Figure 3d). In the IPGTT, DIO rats demonstrated impaired glucose tolerance, with a higher area under the curve (AUC) than the control group (Figure 3d,e). In comparison to control and DR rats, the calculated HOMA-IR revealed insulin resistance in DIO rats (Figure 3g).

DIO rats were characterized by higher BW, BW gain, and body fat mass, while DR rats presented parameters comparable to control rats. Glucose intolerance was found in DIO rats and related to impaired insulin sensitivity as the greater HOMA-IR.

### 3.2. Effects of Short-Term GH Administration on the Circulating Insulin, FFAs, Cholesterol, TG and Liver TG Levels

Short-term administration of GH significantly increased plasma insulin levels in all rats (*p* < 0.05). However, the fasting plasma insulin level of DIO rats was higher than that of control and DR rats (Figure 4a). As shown in Figure 4b, short-term GH administration significantly increased FFA levels only in DR rats (*p* < 0.05). Circulating cholesterol levels as the percentage of the cholesterol level of the control group were decreased after short-term GH treatment in DIO and DR rats (*p* < 0.05) (Figure 4c). For serum TG levels, a significantly decreased TG level was found in the GH-treated DR rats (*p* < 0.05) (Figure 4d). In addition, the short-term GH supplement did not alter the weight of the liver. The percentage of liver weight per body weight is shown in Figure 4e. For liver TG content and liver weight, DIO and DR rats had higher liver TG and liver weight than control rats. Short-term GH administration had no effect on the amount of liver TG (Figure 4f).

These results indicated that short-term GH administration improved the serum lipid profiles in HC-fed rats, in particular, and serum cholesterol levels in both DIO and DR rats. The pronounced effect of the decreased serum TG level was demonstrated in only DR rats. On the other hand, short-term GH treatment did not alter the liver TG level.

### 3.3. Effects of Short-Term GH Administration on mRNA Expression of Genes Related to Hepatic Lipid Metabolism and the Hepatic FAS Protein Levels

In the analysis of the mRNA expression data, only the effect of short-term GH administration was considered. Thus, measurement of the data set was separated for each group. After short-term GH treatment, the mRNA expression of genes linked to hepatic lipid metabolism, particularly FASN and CD36, was significantly reduced in control and DIO rats (FASN; *p* < 0.01, *p* < 0.05, respectively; CD36; *p* < 0.05 in both groups), (Figure 5a,b). FASN and CD36 mRNA expressions also tended to decrease after short-term GH treatment in DR rats (Figure 5c). In addition, LPL mRNA levels were markedly decreased after short-term GH administration in control and DR rats (*p* < 0.01 and *p* < 0.05, respectively). After short-term GH treatment, CPT1A mRNA expression was lowered in control rats (*p* < 0.05), and tended to decrease in DR rats. In contrast, GH markedly increased CPT1A mRNA expression in DIO rats (*p* < 0.05). The VLDLR mRNA expression was significantly increased after short-term GH treatment only in DIO rats (*p* < 0.01). GH administration did not alter the level of Insig2 gene expression in any group. The mRNA expression profiles were shown in Figure 5a–c. In addition, the protein levels of FAS were decreased in all groups after short-term GH treatment, as shown in Figure 5d.

Short-term GH administration downregulated the gene transcription of FASN and CD36 which were related to de novo lipogenesis and fatty acids uptake. These results were in accordance with decreased hepatic FAS levels in all groups after short-term GH treatment. The decreased LPL gene expression reported in control and DR rats after short-term GH treatment indicated that TG hydrolysis and uptake of the fatty acids into the liver were decreased. However, the gene expression of CPT1A, which encoded an enzyme that was responsible for fatty acid oxidation, was upregulated only in DIO rats after GH injection.

### 3.4. Effects of Short-Term GH Treatment on the Expression of Effectors Involved in the GH and Insulin Receptor Signaling Pathways in the Liver

In NSS-treated groups, the JAK2 protein levels in the liver of DIO rats were significantly lower than those of control and DR rats (*p* < 0.05) (Figure 6a). Short-term GH treatment significantly increased JAK2 protein levels in the liver of control rats (*p* < 0.05). In contrast, JAK2 protein expression in the liver of DIO and DR rats was significantly decreased by short-term administration of GH (*p* < 0.001 and *p* < 0.01, respectively). The protein content of STAT5 in DIO rats was significantly lower than that of control and DR rats (*p* < 0.05) (Figure 6b), similar to the trend for JAK2. However, short-term GH administration markedly increased the level of STAT5 only in control rats (*p* < 0.05). There was no difference in the amount of PTEN in the liver of DIO or DR rats when compared to control rats. Interestingly, the concentration of PTEN in DIO rats was significantly higher than in DR rats (*p* < 0.05). GH treatment did not affect the level of PTEN in any groups (Figure 6c). DR rats had significantly higher levels of PI3K p110α protein than control and DIO rats (*p* < 0.001) (Figure 6d). Decreased PI3K p110α levels were only observed in DR rats after short-term GH treatment (*p* < 0.05) (Figure 6d). Short-term GH administration significantly increased AKT level in control rats (*p* < 0.05) (Figure 6e), although a similar trend was seen in DIO and DR rats but did not reach statistical significance. IRS-1 protein levels were increased in short-term GH-treated DIO and DR rats (*p* < 0.001 and *p* < 0.05, respectively) (Figure 6f); however, IRS-1 levels were unchanged by GH treatment in control rats.

These results indicated that impaired GHR signaling pathways were found in HC-fed rats, especially in DIO rats. A decrease of the GHR effector molecule, JAK2, was demonstrated after short-term GH administration in DIO rats. For insulin receptor signaling molecules, increasing IRS-1 was reported in HC-fed rats, both DIO and DR rats. The downstream signaling molecules of the insulin receptor, including PI3K/AKT, were increased in control rats, especially for AKT. As a result, an impaired hepatic insulin receptor signaling pathway after short-term GH treatment was shown in HC-fed rats.

## 4. Discussion

The present study demonstrated that short-term GH administration could lead to increased plasma insulin levels, but not alter the levels of plasma FFAs in DIO and control rats. Therefore, the increased plasma insulin levels may result from GH-stimulated pancreatic islet β-cell growth [40,41]. Serum cholesterol and TG levels were decreased after short-term GH treatment in DIO and DR rats. These effects were due to the upregulation of hepatic VLDLR mRNA expression. Our results suggested that short-term GH administration provided the benefit of lipid-lowering in metabolic disturbance conditions. This effect was similar to fenofibrate, which is the lipid-lowering agent and is approved to treat dyslipidemia [42]. Short-term GH administration might not alter the liver enzyme activity, especially for ALT and AST, as shown by previous studies [29,43]. Furthermore, GH administration has been shown to improve liver enzymes in obese subjects treated with GH for 6 months [44]. Obesity induction with HC diet in rats resulted in NAFLD with increased liver weight [45] and TG levels [46], which are consistent with our findings.

Short-term GH supplementation regulates the gene transcription of lipid metabolism in the liver. Inhibition of hepatic lipogenesis was suggested by the decreased FASN mRNA expression, which has been reported previously [25,47,48]. The decreased hepatic FAS levels have been found in a similar way to that of mRNA expression in all groups. Furthermore, inhibition of CD36 mRNA expression is also consistent with the effect of short-term GH treatment on lipid uptake. This finding is consistent with a previous report [49]. Our current results also indicate that short-term GH administration decreased the gene transcription of hepatic lipid uptake and buildup, as a previous study demonstrated a decrease in LPL gene expression in non-obese conditions [50]. Furthermore, a previous study found that 10 weeks of long-term GH administration in mice with a low-density lipoprotein (LDL) receptor deficiency fed a high fat diet (HFD) resulted in lower liver triglyceride levels and downregulation of CD36 mRNA expression [51]. In our current study, short-term GH administration (3 days) resulted in decreased FASN, CD36, and LPL mRNA expressions but not decreased liver TG levels. Further experiments with the long-term GH administration will be required to confirm the lipid-lowering properties of the liver in DIO rats. VLDLR mRNA expression is another aspect of the effect of short-term GH administration on the regulation of hepatic lipid uptake. Our findings demonstrate that the VLDLR mRNA expression was upregulated in the obese state. There is evidence that VLDLR is upregulated in response to endoplasmic reticulum (ER) stress [52]. However, this effect was interesting to further investigate the underlying mechanisms which could be attributed to fenofibrate, the lipid-lowering agent. Furthermore, the level of CPT1A expression was one of the key molecules mediating hepatic long-chain fatty acid oxidation. The findings showed that GH treatment increased fatty acid oxidation in obese conditions. In non-obese states, however, GH administration reduced fatty acid oxidation. These results indicated that short-term GH administration potentiated hepatic fatty acid oxidation in subjects with metabolic disturbance. It was noted that hepatic CPT1A may be a therapeutic target for gene therapy in obese and NAFLD subjects. Increased CPT1A expression is expected to improve metabolic panels of lipid metabolism and liver function [53].

Aside from lipid metabolism in the liver, GH effects are mediated by a GH receptor-dependent signaling pathway. The GH receptor’s molecular pathways, including JAK2/STAT5, have been linked to insulin sensitivity [54]. JAK2 levels were lower in HC-fed rats, whereas STAT5 levels were lower only in DIO rats (Figure 6a,b). It is worth noting that hyperinsulinemia has been shown to suppress the JAK2/STAT5 signaling pathway [54,55]. We investigated how short-term GH administration affected molecular signaling pathways of GH and insulin receptors. GH increased the hepatic levels of JAK2 and STAT5, which are the downstream effectors of GH receptor signaling, in insulin-sensitive conditions, as demonstrated in control rats. In contrast, HC-fed rats with decreased insulin sensitivity had lower hepatic levels of both JAK2 and STAT5. Furthermore, the effector proteins of the insulin receptor were investigated in this study. Our findings revealed significantly increased levels of IRS-1 only in GH-treated HC-fed rats. The hepatic insulin receptor stimulates the expression of IRS-1 and its downstream effectors, including the PI3K/AKT signaling pathway, which mediates insulin’s metabolic effects [56,57,58,59]. After short-term GH administration, increased levels of AKT were reported in all test groups, but statistical significance was found only in the control group. However, the levels of the catalytic subunit of PI3K, p110α, were altered only in DR rats. It was noted that p110α levels in DR-NSS rats exhibited the highest levels compared to control and DIO rats. Although p110α plays an important role in the metabolic pathways of insulin action in the liver (e.g., maintaining insulin sensitivity and lowering gluconeogenesis), insulin sensitivity is complex and each target organ exhibits unique changes in metabolic pathways [59]. Based on the findings of the higher levels of p110α in the liver of DR rats, it was hypothesized that these p110α levels were connected with the degree of systemic insulin resistance. For instance, DR rats had a lower risk for impaired glucose tolerance when compared to DIO rats. The decrease in liver p110α levels following short-term GH administration in DR rats is consistent with the development of increased insulin sensitivity. Previous research found that lower p110α levels were associated with lower fatty acid uptake to the liver [60], which was linked to higher plasma FFA levels in DR rats. GH primarily affects lipolysis and raises plasma FFA levels [61], which are then taken up by the liver. However, GH might cause a controversial effect, depending on nutritional and metabolic status. Furthermore, a negative regulator of PI3K signaling, PTEN, was unaffected by short-term GH treatment in either HC or control rats. However, an earlier study discovered that chronic GH administration for 3 weeks increased hepatic PTEN levels and was associated with the development of insulin resistance [62]. It should be noted that the action of GH depends on whether the treatment is acute or chronic. We hypothesized that systemic insulin sensitivity was maintained by an early compensatory mechanism. Figure 7 depicts the overall effects of short-term GH administration.

The roles of insulin and GH related to the pathophysiology of NAFLD progression are complex. According to our findings, short-term GH administration may provide evidence of decreased liver lipid uptake and promote lipid metabolism by altering the mRNA expression. Furthermore, our findings indicate that the different stages of metabolic disturbance in DR and DIO rats have a significant impact on the responses to short-term GH administration. Despite similar hepatic TG levels in NSS and GH treated groups, GH possesses the ability to improve lipid metabolism by transcriptional regulation. In conclusion, short-term GH administration in the obese condition promotes beneficial effects on the hepatic lipid metabolism, which may slow the progression of NAFLD.

## 5. Conclusions

This study demonstrated that short-term GH treatment affected the gene transcription of liver lipid metabolism in obese rats. Improving lipid metabolism involved decreasing lipid acquisition (the gene transcription levels of fatty acid uptake and de novo lipogenesis and the hepatic FAS protein levels) and increasing lipid elimination (the gene transcription levels of fatty acids oxidation). These findings supported the transcriptional regulation of GH for further study in the therapeutic approach to NAFLD.

## Figures and Tables

**Figure 1 biomedicines-11-01050-f001:**
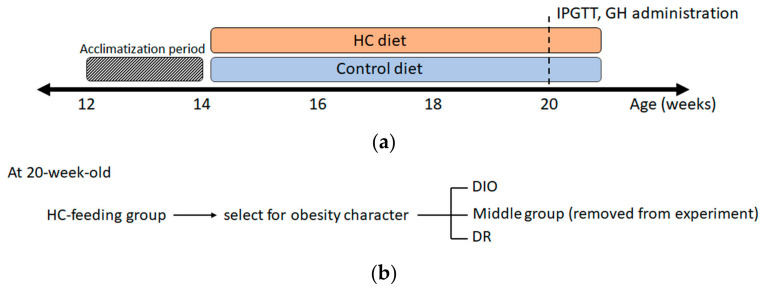
(**a**) The experiment timetable. (**b**) The HC-feeding rat groups after 6-weeks of dietary intervention.

**Figure 2 biomedicines-11-01050-f002:**
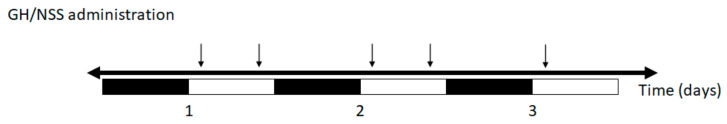
The rats in each group were treated with either GH or NSS for 3 days (short-term administration). The arrows indicate GH or NSS administration (0800h and 1600h). The black and white bars show dark and light phases, respectively (12 h/12 h).

**Figure 3 biomedicines-11-01050-f003:**
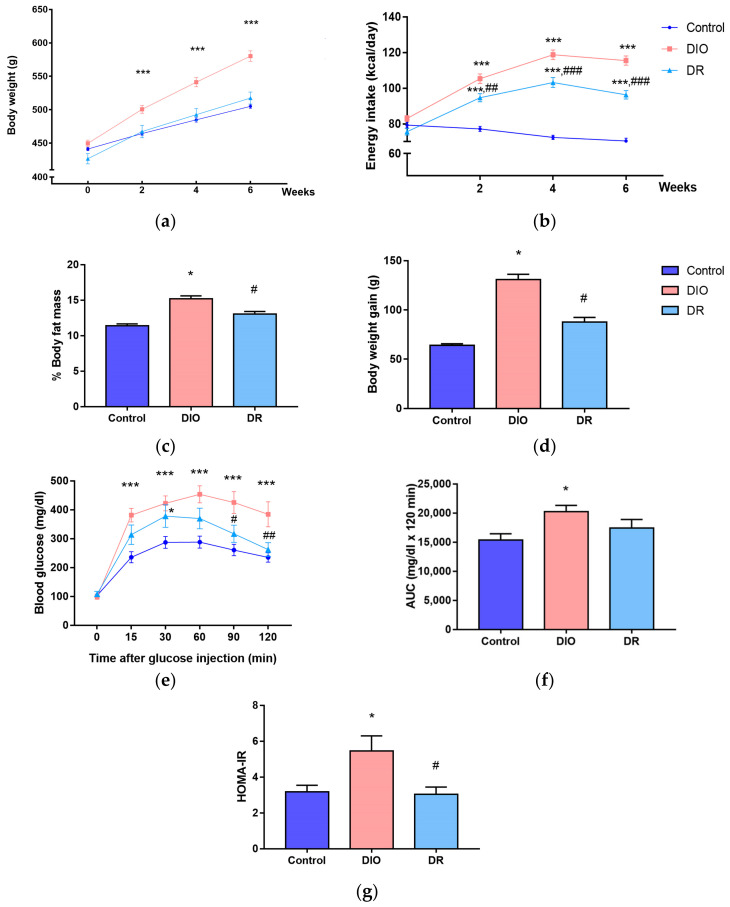
(**a**) The body weight (BW) of rats during diet manipulation for 6 weeks; *** represents statistical significance compared to other groups (*p* < 0.001). (**b**) Energy intake during the 6-week period; *** represents statistical significance when compared to other groups (*p* < 0.001), ## represents statistical significance when compared to DIO rats (*p* < 0.01), ### represents statistical significance when compared to DIO rats (*p* < 0.001). (**c**) The percentage of body fat mass in each group; * represents statistical significance when compared to the control group (*p* < 0.05), # represents statistical significance when compared to DIO rats (*p* < 0.05). (**d**) The body weight gain during 6 weeks of diet manipulation; * represents statistical significance compared to the control group (*p* < 0.05), # represents statistical significance when compared to DIO rats (*p* < 0.05). (**e**) Intraperitoneal glucose tolerance test (IPGTT) in control, DIO and DR rats; *, *** represents statistical significance when compared to the control group (*p* < 0.05, *p* < 0.001, respectively), #, ## represents statistical significance when compared to DIO rats (*p* < 0.05, *p* < 0.01, respectively). (**f**) Area under the curve (AUC) of IPGTT; * represents statistical significance when compared to the control group (*p* < 0.05). (**g**) The calculated homeostasis model of insulin resistance (HOMA-IR) in control, DIO and DR rats; * represents statistical significance when compared to the control group (*p* < 0.05), # represents statistical significance when compared to DIO rats (*p* < 0.05). All results are presented as mean ± SEM; *n* = 12 per group (control and DIO rats), *n* = 10 per group (DR rats).

**Figure 4 biomedicines-11-01050-f004:**
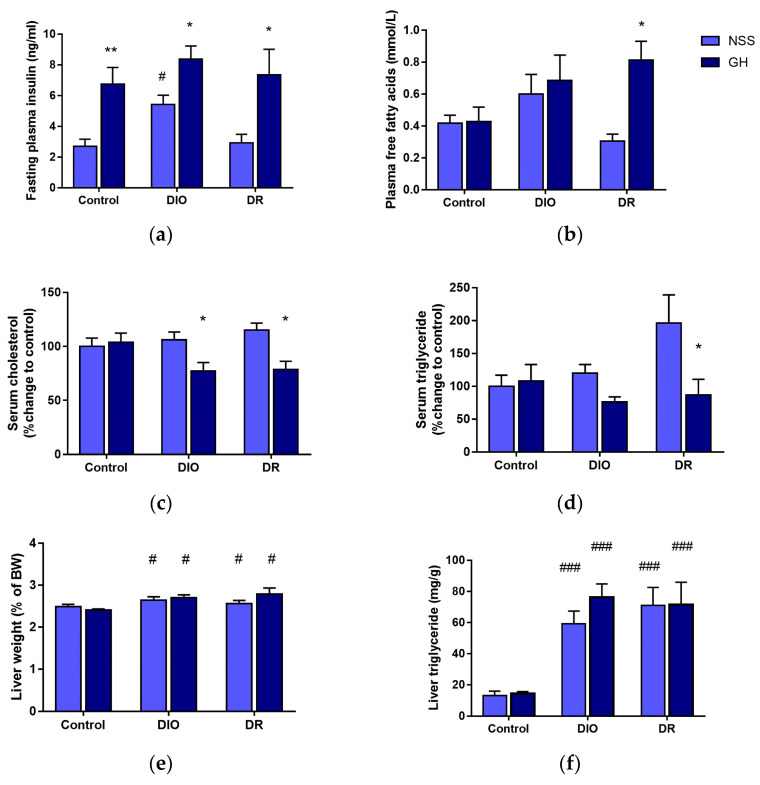
(**a**) Plasma insulin after short-term GH administration; *, ** represent statistical significance between GH and NSS groups (*p* < 0.05, *p* < 0.01, respectively), # represents statistical significance when compared to the control group (*p* < 0.05). (**b**) Plasma free fatty acids after short-term GH administration; * represents statistical significance between GH and NSS groups (*p* < 0.05). (**c**) Serum cholesterol levels (% change to the control group) after short-term GH treatment; * represents statistical significance between GH and NSS groups (*p* < 0.05). (**d**) Serum triglyceride levels (% change to the control group) after short-term GH treatment; * represents statistical significance between GH and NSS groups (*p* < 0.05). (**e**) The percentage of liver weight in each group; # represents statistical significance when compared to the control group (*p* < 0.05). (**f**) Liver triglyceride concentration (mg/g tissue) in each group; ### represents statistical significance when compared to the control group (*p* < 0.001). All results are presented as mean ± SEM; *n* = 6 per experimental group (NSS/GH of control and DIO rats), *n* = 5 per experimental group (NSS/GH of DR rats).

**Figure 5 biomedicines-11-01050-f005:**
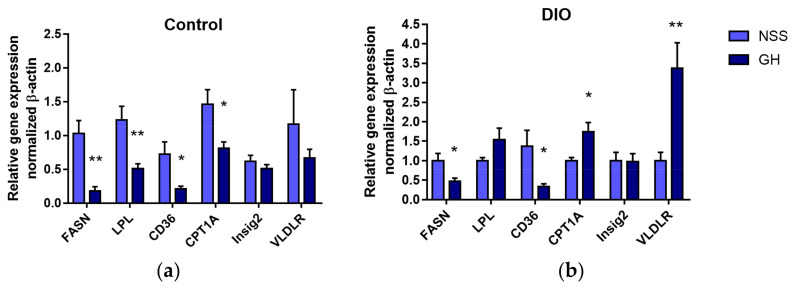
(**a**) The liver mRNA expression in the control group; *, ** represents statistical significance between GH and NSS groups (*p* < 0.05, *p* < 0.01, respectively). (**b**) The liver mRNA expression in the DIO group; * represents statistical significance between GH and NSS groups (*p* < 0.05). (**c**) The liver mRNA expression in the DR group; *, ** represents statistical significance between GH and NSS groups (*p* < 0.05, *p* < 0.01, respectively). (**d**) Liver FAS protein levels; * represent statistical significance differences between GH and NSS groups (*p* < 0.05). All results are presented as mean ± SEM; *n* = 6 per experimental group (NSS/GH of control and DIO rats), *n* = 5 per experimental group (NSS/GH of DR rats).

**Figure 6 biomedicines-11-01050-f006:**
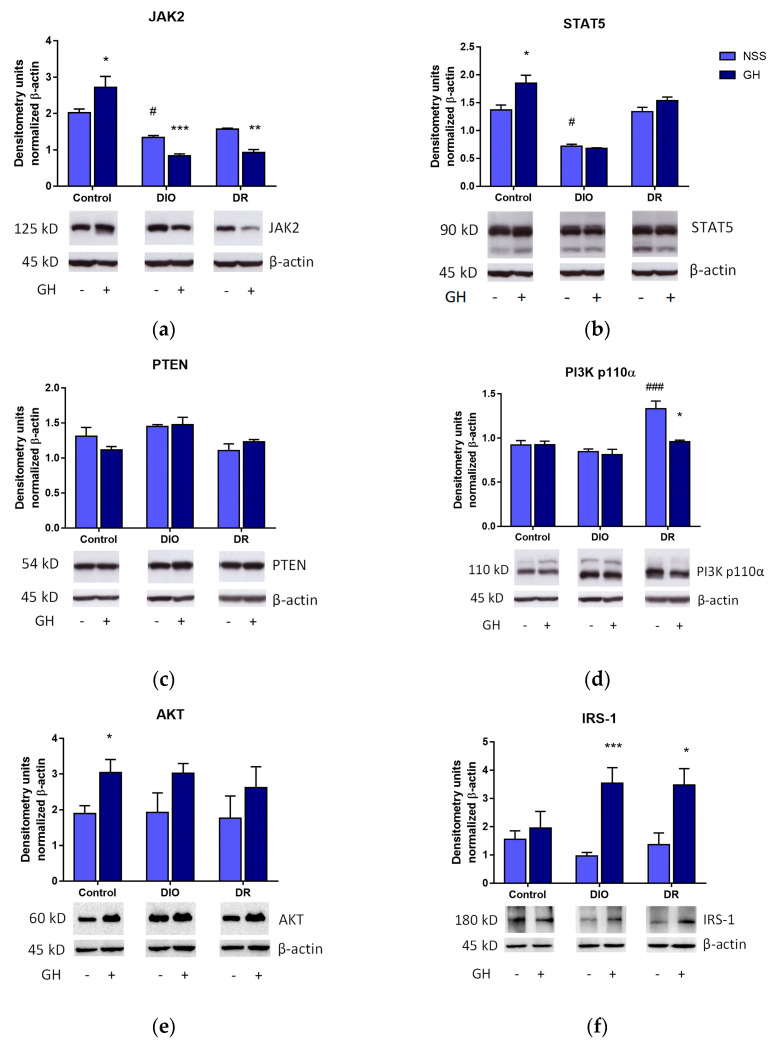
(**a**) Liver JAK2 protein levels; *, **, *** represent statistical significance differences between GH and NSS groups (*p* < 0.05, *p* < 0.01, *p* < 0.001, respectively), # represents statistical significance differences with the control group (*p* < 0.05). (**b**) Liver STAT5 protein levels; * represents statistical significance differences between GH and NSS groups (*p* < 0.05), # represents statistical significance differences with the control group (*p* < 0.05). (**c**) Liver PTEN protein levels. (**d**) Liver PI3K p110α protein levels; * represents statistical significance differences between GH and NSS groups (*p* < 0.05), ### represents statistical significance differences with the control group (*p* < 0.001). (**e**) Liver AKT protein levels; * represents statistical significance differences between GH and NSS groups (*p* < 0.05). (**f**) Liver IRS-1 protein levels; *, *** represent statistical significance differences between GH and NSS groups (*p* < 0.05, *p* < 0.001, respectively). All results are presented as mean ± SEM; *n* = 6 per experimental group (NSS/GH of control and DIO rats), *n* = 5 per experimental group (NSS/GH of DR rats).

**Figure 7 biomedicines-11-01050-f007:**
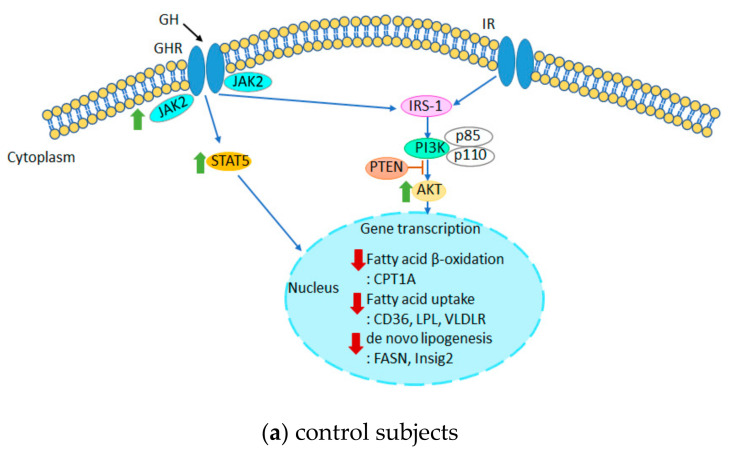
(**a**) The effect of short-term GH administration on the GHR and IR signaling pathway and gene transcription of lipid metabolism in hepatocytes of control rats. (**b**) The effect of short-term GH administration on the GHR and IR signaling pathway and gene transcription of lipid metabolism in hepatocytes of DIO rats. ↑ = increase; ↓ = decrease.

**Table 1 biomedicines-11-01050-t001:** The sequence primers for quantitative RT-PCR.

Fatty acid synthase (FASN)	Forward (5′→3′) Reverse (5′→3′)	GCATTTCCACAACCCCAACC AACGAGTTGATGCCCACGAT
Lipoprotein lipase (LPL)	Forward (5′→3′) Reverse (5′→3′)	ATGGCACAGTGGCTGAAAGT CCGGCTTTCACTCGGATCTT
Cluster of differentiation 36 (CD36)	Forward (5′→3′) Reverse (5′→3′)	TGGACTTGTACTCTCTCCTCGG TCGTGCAGCAGAATCAAGGA
Carnitine palmitoyltransferase 1A (CPT1A)	Forward (5′→3′) Reverse (5′→3′)	TGCAGAGCAATAGGTCCCC ACACCCACCACCACCACGATAAG
Insulin induced gene 2 (Insig2)	Forward (5′→3′) Reverse (5′→3′)	GCGTGTTCCTGGCTTTAGTG CGACTTTAGCACTGGCGTGA
Very low-density lipoprotein receptor (VLDLR)	Forward (5′→3′) Reverse (5′→3′)	GTGATGAGCTGGACTGTGCT GCCACACTGCTCAAGAGACT
β-actin	Forward (5′→3′) Reverse (5′→3′)	CCACCATGTACCCAGGCATT AGGGTGTAAAACGCAGCTCA

## Data Availability

The data presented in this study are available on request from the corresponding author.

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
