# Peer review of "Short-Term Growth Hormone Administration Mediates Hepatic Fatty Acid Uptake and De Novo Lipogenesis Gene Expression in Obese Rats"

_biomedicines, 2023, doi:10.3390/biomedicines11041050_

Round 1

Reviewer 1 Report

This study with the title: Short-Term Growth Hormone Administration Reduces Hepatic Fatty Acid Uptake and De Novo Lipogenesis in Obese Rats” by  Sutharinee Likitnukul et al. analyzed the effect of short-term growth hormone injection in rats fed a control diet, diet-induced obese rats and obesity resistant effects.

These findings indicated that short-term GH administration reduces hepatic fatty acid uptake and lipogenesis, also promoted fatty acid oxidation in DIO rats. “ this can not be concluded because hepatic TGs did not change and protein levels of e.g. FAS were not analyzed.

It is in addition unlikely that short-term injection of GH will improve NAFLD. No effect on liver TGs and higher insulin will not improve NAFLD.

How is FAS regulated? Is there an effect of insulin / STAT5 described in the literature?

Why was VLDLR and not LDLR analyzed?

The mRNA data shown in figure 5 have to be verified at the protein level at least for FAS and CD36.

increased plasma insulin levels, resulting in systemic insulin resistance.” This is not correct. Higher insulin is a result of insulin resistance.

An intraperitoneal glucose tolerance test (IPGTT) was performed to measure systemic insulin sensitivity.” This assay measures glucose deposition but not necessarily insulin sensitivity. For this assay insulin has to be injected.

The original figure file shows the figures in the manuscript but not the original blots.

The discussion is confusing and has to be rewritten for clarity. The manuscript has to be corrected by a native speaker.

In insulin-sensitive subjects, GH increased the hepatic levels of JAK2 and STAT5, which are the downstream effectors of GH receptor signaling.” Rats?

GH signaling seems to be impaired in the obese but still has significant effects. This should be more precisely shown in figure 7.

 “Taken together, our results suggest  that short-term GH supplementation improves liver lipid metabolism and may slow the progression of NAFLD.”

Liver triglycerides do not change and further lipids were not measured in the liver. This conclusion is not correct.

Reviewer 2 Report

Growth hormone (GH) plays an important role in the regulation of metabolism, the effect of GH on insulin resistance with long-term treatment of GH has been reported, while short-term treatment with GH on insulin sensitivity and the underlying mechanism is still unclear. In this study, Sutharinee Likitnukul et al using Diet induce obesity(DIO) rat model investigated the short-term treatment of GH in vivo, data suggested that Short-Term Growth Hormone Administration Reduces Hepatic 2 Fatty Acid Uptake and De Novo Lipogenesis in DIO Rat, however, some of the data still not convincing enough and further experimental needed for the further consideration. Several comments and suggestions are below:

1. In Figure 6, given the individual difference for the in vivo study, The WB should be including more samples for each group, in addition, the samples should be loaded in the same gel. And the number used should be provided in the figure legend. Lastly, the phosphorylation of the GH and insulin signaling pathway should include as well.

2. The Morphology of the liver tissue and related organs should be further performed by Hematoxylin and eosin stain.

3. In figure 4, the author mentioned GH increased the insulin level, Does short-term GH affect gluconeogenesis in the Liver?

4. As the title said that Short-Term Growth Hormone Administration Reduces Hepatic 2 Fatty Acid Uptake and De Novo Lipogenesis in DIO Rat, Does GH short-term treatment improvement in the liver reflected by GTT or ITT?

5. As a suggestion, its not necessary to include the DR groups data in the manuscript if this group's resistance to DIO and consider normal chow. And the mRNA level in figure 5 should combine the group Control and DIO group for better comparison.

6. The sub-title and results part should provide some conclusion or summary instead of the description of the experiment design only.

7. The text of 3.3 mentioned ”LPL mRNA levels were also reduced in DR rats, but the difference was not statistically significant”, This is not consistent with figure 5C since the figure shows a significant label with *.

8. Figure 4D DIO group shows a significantly decreased TG level in serum?

9. The methods indicated GH treatment by subcutaneous injection twice every day for 3 days. What is the specific reason using subcutaneous injection for twice per day instead of intraperitoneal once per day?

Round 2

Reviewer 1 Report

The discussion still has to be greatly improved.  There are still grammatical errors.

Sentence such as:“and promote lipid metabolism by altering the  mRNA expression.” Are not correct because regulating mRNA expression alone will most likely not change a lot. This and related unclarities have to be corrected.

It is unclear why the authors do not show the original blots as is required by the journal.

There is a very strong effect of GH on plasma free fatty acids and triglycerides only in DR rats.  This is neither explained nor discussed.

Moreover, how do the authors explain the lower serum cholesterol in DIO and DR rats?

It is unclear why the authors do not provide a FAS immunoblot. The protein lysates are available and performing three immunoblots is not so much work.

Reviewer 2 Report

The manuscript improved. However. A couple of minor revisions are needed.

1. As the author mentioned In Figure 6 multiple samples were used and loaded in the same gel for the WB, authors should provide the uncropped gel images instead of the representative images. And the phosphorylated form of JAK2, STAT5, AKT, and IRS-1 images the author should provide is not readable without the labeling.

2. Given the TG not changed in the Liver tissue between the groups. The H&E staining from the Liver tissue not required the frozen section. and the paraffin section works well to check the lipids droplets.

Round 3

Reviewer 1 Report

The manuscript was revised accordingly